# Effects of Heat-Killed *Lactobacillus plantarum* L-137 Supplementation on Growth Performance, Blood Profiles, Intestinal Morphology, and Immune Gene Expression in Pigs

**DOI:** 10.3390/vetsci10020087

**Published:** 2023-01-24

**Authors:** Wandee Tartrakoon, Rangsun Charoensook, Tossaporn Incharoen, Sonthaya Numthuam, Thitima Pechrkong, Satoru Onoda, Gaku Shoji, Bertram Brenig

**Affiliations:** 1Division of Animal Science and Feed Technology, Department of Agricultural Sciences, Faculty of Agriculture, Natural Resources and Environment, Naresuan University, Phitsanulok 65000, Thailand; 2Center of Excellence for Agricultural and Livestock Innovations, Faculty of Agriculture, Natural Resources and Environment, Naresuan University, Phitsanulok 65000, Thailand; 3House Wellness Foods Corporation, 3-20 Imoji, Itami 664-0011, Japan; 4Institute of Veterinary Medicine, Faculty of Agricultural Sciences, Georg-August-University of Göttingen, 37077 Göttingen, Germany

**Keywords:** cytokine gene expression, growth performance, HK L-137, immune response, swine

## Abstract

**Simple Summary:**

Antibiotic growth promoters have long been used in pig diets to maintain gut health and improve growth performance. However, the overuse of antibiotics has led to the emergence of antibiotic-resistance microbes, an imbalance of the healthy intestinal microflora, and antibiotic residues. As a result, probiotic bacteria have been used as an alternative to antibiotics in pig production. Nevertheless, in-feed probiotics may be inconsistent due to differences in preparation methods, feed storage, and ability to survive through the gastrointestinal tract. Heat-killed *Lactobacillus plantarum* L-137 (HK L-137), also known as a paraprobiotic, has been shown to have beneficial effects on the immune response and has no limitations. Our study indicates that supplementing HK L-137 in pig diets can improve production performance and promote immune function.

**Abstract:**

In the present study, the effects of dietary heat-killed *Lactobacillus plantarum* L-137 (HK L-137) on the productive performance, intestinal morphology, and cytokine gene expression of suckling-to-fattening pigs were investigated. A total of 100 suckling pigs [(Large White × Landrace) × Duroc; 4.5 ± 0.54 kg initial body weight (BW)] were used and assigned to each of the four dietary treatments as follows: (1) a control diet with antibiotics as a growth promoter (AGP) from the suckling phase to the grower phase and no supplement in the finisher phases; (2) a control diet without antibiotics as a growth promoter (NAGP); (3) a control diet with HK L-137 at 20 mg/kg from the suckling phase to the starter phase and no supplement from the grower phase to the finisher phases (HKL1); and (4) a control diet with HK L-137 at 20 mg/kg from the suckling phase to the weaner phase, at 4 mg/kg from the starter phase to the finisher 1 phase, and no supplement in the finisher 2 phase (HKL2). During the weaner–starter period, the pigs fed on the AGP and HKL2 diets showed significantly higher weight gain and average daily gain (ADG) than those in the NAGP group (*p* < 0.05). The pigs in the AGP, HKL1, and HKL2 groups showed greater ADG than those in the NAGP groups (*p* < 0.05) throughout the grower–finisher period. The suckling pigs in the HKL1 and HKL2 groups showed a higher platelet count (484,500 and 575,750) than in the others (*p* < 0.05); however, there were no significant differences in the other hematological parameters among the treatment groups. The relative mRNA expression level of *IFN- ß* of the suckling and starter pigs were significantly higher in the HKL1 and HKL2 groups than in the others (*p* < 0.05), while the *IFN-γ* showed the highest level in the HKL2 suckling pigs (*p* < 0.05). These results demonstrate that a HK L-137 supplementation could stimulate the immune response in suckling and starter pigs and promote the growth performance in finishing pigs.

## 1. Introduction

The weaning of piglets is acutely stressful during the initial stage of life, causing a reduction in feed intake, increased incidence of diarrhea, growth retardation, susceptibility to pathogens, and increased mortality [1,2]. These influences may have an effect on pig performance throughout the finishing stage; thus, antibiotic growth promoters (AGPs) have been widely used in weaning pig diets to maintain gut health and increase growth performance [2,3]. However, the continuing use of antibiotics in animal feed has resulted in problems; for example, antibiotic resistance in bacteria has emerged, along with an imbalance of the healthy intestinal microflora and antibiotic residues in animal products [3]. As a result, several countries have banned or restricted the use of AGPs [4,5]. Therefore, there is an ongoing search for non-antibacterial growth promoters that act in vivo, are fast acting, possess a broad spectrum of activity, and can subsequently promote the growth performance of pigs [6]. Due to the aforementioned problems, livestock producers are looking for alternative natural supplements with aspects that can enhance physiological functions and activate animal immune responses. Famous natural and widely used growth promoters are probiotic bacteria [7,8].

Probiotics are live microorganisms that will benefit the host when administered appropriately [9,10]. Health-promoting probiotics contribute to healthy gut microbiota and activate the host immune response, thus positively enhancing growth performances. Lactic acid bacteria (LAB), which are non-pathogenic Gram-positive inhabitants of conventional human and animal intestines, are identified as being associated with health-promoting results by facilitating non-specific immune system enhancement and safety against intestinal infection [11]. Previous research on the use of dietary supplementation with probiotics has improved piglets’ immune response [12,13,14,15,16]. Nevertheless, the effects of probiotic supplementation may be inconsistent due to the differences in preparation methods, feed storage, ability to survive passage through the gastrointestinal tract, and antibiotic-resistance gene transfer to other microorganisms [8,17,18,19,20].

Recently, metabolic by-products, dead microorganisms, or other microbial-based non-viable products (for example, yeast cell walls) are being extensively used as a non-antibiotic nutritional approach to improving the productive performance, gut function, and immune system. Previous studies have reported that non-viable microbes, referred to as immunobiotics or paraprobiotics, also exhibit beneficial effects on the immune response that are equivalent to or greater than probiotics. Moreover, they still have no limitations on the processing, storage, and ability to survive in the gastrointestinal tract [21,22,23,24].

Heat-killed *Lactobacillus plantarum* strain L-137 (HK L-137), a strain isolated from fermented foods, is a non-viable, heat-treated *Lactobacillus plantarum* that is resistant to high temperatures in feed processing. It plays a role as an immunostimulant and a growth promoter. HK L-137 has been studied in many land animals, such as mice [21,22,25], broiler chickens [8,17], pigs [26], and aquatic animals, such as kuruma shrimp [27], red sea beam [18,20], and amberjack [19]. These studies indicate that HK L-137 may activate intestinal function, regulate immunological response, and cause an increase in productive performance during the supplementation time. However, its effects on suckling-to-fattening pigs have not been completely investigated [26]. Our study aimed to evaluate the effects of dietary supplementation of HK L-137 on growth performance, hematological parameters, morphology of the small intestine, and expression of cytokine-encoding genes in pigs from the suckling to the finishing stages.

## 2. Materials and Methods

The current study was carried out in the research farm of the Faculty of Agriculture, Natural Resources and Environment, Naresuan University, Thailand. The experimental procedure was approved by the Naresuan University Animal Care and Use Committee (NUACUC; reference no. NU-AG600613).

### 2.1. Preparation of Heat-Killed Lactobacillus Plantarum L-137

Heat-killed *Lactobacillus plantarum* L-137 (HK L-137) was prepared based on the method previously described [21]. In this study, LP Pro^TM^ (House Wellness Foods Corp, Itami, Japan) that contained 10% of HK L-137 and 90% of whey protein, dextrin, and sunflower lecithin was used. It contained 1 × 10^11^ cfu/g of *Lactobacillus plantarum* in the dry product stored at room temperature (information from product instruction).

### 2.2. Experimental Design and Pig Management

Crossbred [(Large White × Landrace) × Duroc; 4.5 ± 0.54 kg initial body weight (BW)] suckling pigs were used and assigned to each of the 4 dietary treatments, with piglets from 10 lactating sows per treatment (Landrace × Large White, 250 ± 50 kg of BW) at a parity of 2–6; the sows gave birth at full term and had the same litter size (more than ten live piglets per litter). The experimental feeds (Creep feed) were offered from 14 days (suckling pigs) to 24 days during the weaning age. The number of piglets per treatment in this period was 118, 133, 117, and 125, respectively.

After that, a total of 100 (castrated males) weaning pigs were selected and arranged from the experimental suckling pigs in each treatment using a completely randomized design, with 25 pigs per treatment (5 pigs per replicate and 5 replicates per treatment). They were separated into the weaning pig phase (24–52 days) and starting pig phase (52–66 days). The pigs were weighed on arrival and every week during the weaner-to-starter phase. The ambient temperature was maintained at 30 °C for the first week after weaning, dropped by 1 °C each week after that, and maintained at 26–28 °C. All pigs were given ad libitum feed and water. The pigs from the starting pig phase were transferred to the growing–finishing pig house and weighed every two weeks during the growing–finishing pig phase. The ambient temperature was maintained at 26–28 °C, and the humidity was controlled between 60 and 70%. All pigs were given ad libitum feed and water.

### 2.3. Dietary Treatments

All nutrients of the dietary treatment met or exceeded the NRC requirements [28]. The formula and chemical composition of the experimental diet are presented in Table 1 as follows: (1) a control diet with antibiotics as a growth promoter (AGP) (amoxicillin and colistin) at 300 mg/kg from the suckling phase to the grower phase and no supplement during the finisher phases; (2) a control diet without antibiotics as a growth promoter (NAGP); (3) a control diet with HK L-137 at 20 mg/kg from the suckling phase to the starter phase and no supplement from the grower to the finisher phases (HKL1); and (4) a control diet with HK L-137 at 20 mg/kg from the suckling phase to the weaner phase, at 4 mg/kg from the starter phase to the finisher 1 phase, and no supplement during the finisher 2 phase (HKL2).

### 2.4. Sample Collection

For sample collection, ten pigs from each group (two pigs per replication) at 24 days of age and at 66 days of age (starter pigs), with live weights close to the group average, were randomly selected. EDTA-blood samples were collected [29], and all blood parameters were analyzed using an Abbott Cell-Dyn 3700 hematological analyzer (GMI, Ramsey, MN, USA). Finally, all pigs were euthanized to collect the entire small intestine for gut morphology analysis and spleen for gene expression analysis.

### 2.5. Microscopic Gut Morphology

Each intestinal specimen of about 2–3 cm length was transversely cut from the midpoint of the duodenum, jejunum, and ileum, and gently flushed with 0.1 M phosphate buffered saline (PBS). The intestine was divided into segments based on the length: duodenum (first 10%), jejunum (middle 75%), and ileum (final 15%) [30]. Then, the washed specimens were immersed in buffered formalin and kept in a refrigerator at 4 °C. The samples were dehydrated consecutively with 70, 80, 90, and 100% (absolute) ethanol. An intestinal specimen was embedded in paraffin wax and cut into thin slices of 5 µm thickness. Finally, all sample slides were studied using a compound microscope equipped with a digital camera and an OPTIKA Vision lite software (OPTIKA B-380 series, OPTIKA Microscopes, Bergamo, Italy). The variables of intestinal morphology included villus height, villus area, crypt depth, and villus height per crypt depth. These measurements were analyzed separately for each intestinal segment, according to Incharoen et al. [8].

### 2.6. Quantitative RT-PCR Analysis

After sample collection, total RNA was extracted from the splenic tissue using the RiboZolTM RNA solution (VWR Life Science, PA, USA) and a tissue lysis buffer, according to the manufacturer’s instructions. Each sample was diluted and used in a reverse transcriptase-polymerase chain reaction (qRT-PCR). The resulting cDNA samples were divided into aliquots and subjected to a real-time polymerase chain reaction (real-time PCR). The primers used in the semi-quantification of cytokine expression, i.e., *IL-12*, *IFN-β*, *IFN-γ*, and reference gene (*ß-actin*), were designed based on the gene sequences deposited at https://www.ncbi.nlm.nih.gov (accessed on 8 March 2021) (Table 2). The real-time PCR was performed using the fluorescent dye LightCycler® 480 SYBR Green I Master Mix in the LightCycler machine (Real-Time PCR System; Roche Life Science, PA, USA). The ΔCt values were calculated by subtracting the experimental Ct values from the Ct values for the housekeeping gene targets amplified within each sample and were normalized with the ß-actin gene. A relative mRNA expression analysis was carried out using the 2^−∆∆Ct^ method to account for the exponential amplification of the PCR products, as described in our previous study [8].

### 2.7. Statistical Analysis

The various parameters tested for the dietary treatments were compared using a one-way analysis of variance (ANOVA), followed by a multiple comparison test. Each pen represented the experimental unit for growth performance traits, while each pig was examined for hematology, intestinal morphology, and gene expression. The data were presented as means ± SEM. Significant differences between the treatment means were determined using the Duncan’s multiple-range test. The significance of the difference was defined at *p* < 0.05. All statistical analyses were conducted using the statistical software SPSS, version 17.0 (SPSS Inc., Chicago, USA) [31].

## 3. Results

### 3.1. Growth Performances

In the randomly selected litters of all treatment groups, no significant differences (*p* > 0.05) were observed in the litter size of both the total born and live-born between the groups (Table 3). There were no significant differences (*p* > 0.05) in piglet BW, WG, ADG, ADFI, G:F, and incidence of diarrhea between the groups of pigs during the pre-challenge (birth to 14 days old), post-challenge (14 to 24 days old), and overall suckling period (birth to 24 days old).

The effects of the dietary supplementation of HK L-137 on the growth performances of the pigs from the weaner to the starter phases are presented in Table 4. Our study showed that BW, WG, ADG, ADFI, and G:F were similar in all treatment groups at 24–38 days, at 52–66 days, and during the entire 66-day experimental period. However, from days 38 to 52 post-weaning, WG and ADG were greater (*p* < 0.05) in the AGP and HKL2 groups compared to the NAGP group. Table 5 shows the effects of the supplementation of HKL1 and HKL2 on the growth performance of the pigs from the grower to finisher phases, compared to the AGP and NAGP groups. During the grower phase (25–50 kg BW), we observed no significant differences in ADG, ADFI, and G:F between the treatment groups. However, the experimental period in the AGP group was the shortest (*p* < 0.05), with no significant difference from the HKL1 group.

The experimental period of the NAGP group was the longest (*p* < 0.05) compared to the other groups, with no significant differences from the HKL2 group. There were no significant differences in any growth performance parameters among the treatment groups during the finisher 1 phase (50–75 kg BW) and the finisher 2 phase (75–100 kg BW). An exception was that the best G:F (*p* < 0.05) was observed in the HKL2 group, with no significant differences compared to the AGP group in the pigs during the finisher 2 phase. The growth performance throughout the grower–finisher phase (25–100 kg BW) of the pigs did not significantly differ in terms of ADFI and G:F between the treatment groups. However, the pigs in the AGP, HKL1, and HKL2 groups showed higher ADG than the pigs in the NAGP groups (*p* < 0.05). It was also observed that the group of pigs supplemented with HK L-137 (HKL1 and HKL2) and AGP took less feeding period to attain a BW of 100 kg from 25 kg, compared to the pigs in the NAGP group (*p* < 0.05).

### 3.2. Hematological Parameters and Intestinal Morphology

There was no effect of the dietary HK L-137 supplementation on the suckling pigs’ hematological parameters until weaning (*p* > 0.05), except that the platelet counts in the HKL1 and HKL2 groups were significantly higher (*p* < 0.003) than the AGP and NAGP groups. However, no differences in the hematological parameters of the suckling pigs were observed between the treatment groups (Table 6).

The morphological measurements of the duodenal, jejunal, and ileal mucosae in the suckling pigs and the weaning pigs fed with HK l-137 are shown in Table 7. There are no significant differences in the villus height, crypt depth, villus height/crypt depth ratio, and villus area among the treatment groups.

### 3.3. Expression of Cytokine-Encoding Genes in Pig Spleen

The spleen acts as a filter for blood as part of the immune system. To determine the effects of HK L-137 on cytokine production, we evaluated the expression of interferon (*IFN-γ* and *IFN-β*) and interleukin (*IL-12β*) genes in the spleen of the pigs at the end of the suckling phase (24 days old) and at the end of the starter phase (73 days old). The mean values of relative mRNA expression are presented in Figure 1. At 24 days of age, the HKL2 group showed the highest relative RNA expression level of *IFN-γ* compared to the other groups. On the other hand, in both the 24- and 73-day-old pigs, the HKL1 and HKL2 groups showed a higher relative expression of *IFN-β* than the AGP and NAGP groups (*p* < 0.05). However, there was no significant difference in the expression of *IL-12* at 24 and 73 days of age (*p* > 0.05).

## 4. Discussion

Weaning pigs usually have poor growth performance, and their gastrointestinal tract is susceptible to infections that can readily cause post-weaning diarrhea. This effect may have an impact on pig performance throughout the finishing stage. Therefore, antibiotic growth promoters (AGPs) are used in pig diets to support gut function and enhance growth performance [2,3]. Probiotic bacteria have been employed as an alternative to antibiotics in pig production. Nonetheless, in-feed probiotics may be uneven due to inconsistencies in preparation methods, feed storage, and ability to survive the gastrointestinal tract [18,19,20]. Previous studies have shown that heat-killed probiotics, which can be obtained by the heat-treated method, can complement probiotics’ risk and stability and exhibit beneficial effects that are equal to or better than living ones [8,21,22]. Moreover, heat-killed probiotics may release exopolysaccharides (EPS), lipoteichoic acids (LTA), and other components with immune-regulatory actions against pathogens [21,22,24]. However, the effect of heat-killed probiotics in suckling-to-fattening pigs has remained unclear [26].

In this study, the effects of dietary supplementation of heat-killed *Lactobacillus plantarum* strain L-137 (HK L-137) were investigated concerning the growth performance, hematological parameters, intestinal morphology, and expression of cytokine-encoding genes in pigs from the suckling to the finisher phases. Such variations in pig development and growth are probably the result of adaptations to various supplements, farm hygiene, dietary composition, and feed forms [6,32,33,34]. Our results indicated increased weight gain and ADG of weaning pigs that were fed dietary HK L-137 at both tested levels, compared to the NAGP group, for 38–52-day-old pigs. These results for pigs at the 38–52-day stage were in accordance with other studies that reported an enhancement in the growth performance of pigs that were fed diets supplemented with *Lactobacillus* spp. [32,33,34]. As growing–finishing pigs have a mature gastrointestinal tract with higher digestive enzyme activity, immune capacity, and disease resistance, the influence of HK L-137 in the grower–finisher pigs in this study might be relatively limited. However, our results indicated that the dietary HK L-137 in the treatment HKL groups significantly improved feed efficiency during the finisher phase. Furthermore, both the HKL1 and HKL2 groups showed improved ADG throughout the grower–finisher phase (25–100 kg BW), which was similar to the AGP group.

One possible explanation for this finding is that small changes in FE and ADG during the weaner–starter and grower–finisher stages accumulated up to a point of showing a significant difference across the trial period. These findings agree with the results of Kim et al. [6,35]. They observed that bacteriophages and probiotics could improve different performance parameters, and bacteriophages had a more pronounced effect than probiotics. Moreover, Balasubramanian et al. [36] indicated that supplementation with *Bacillus* spp. probiotics had a linear correlation with ADG and G:F at week 16 and had a significant linear effect on ADG and G:F in the overall experiment, with no differences observed on ADFI between the dietary treatments during the entire period. Comparably, Meng et al. [37] reported a rise in ADG following a dietary supplementation of *Bacillus subtilis* and *Clostridium butyricum* endospores during the experiment. However, G:F was not enhanced throughout the finisher phase. Furthermore, Shon et al. [38] did not find a difference in the growth performance of grower–finisher pigs fed with a diet supplemented with 0.2% probiotics containing *L. reuteri*, *L. salivarius*, *L. plantarum*, and a yeast complex. Although previous studies have reported that live or heat-killed *Lactobacillus* strains provide a functional component that supports nutrient utilization and immune function, leading to improved growth performance in weaning pigs [26,33,34,39], our findings suggest that supplementing HK L-137 during the suckling to fattening stages can promote the growth performance of the finisher pigs.

Blood hematological parameters can reveal information about body metabolism and health. In this study, the suckling and starter pigs did not show statistically significant differences among the treatment groups, with the exception that the suckling pigs in the groups supplemented with HK L-137 (HKL1 and HKL2) having a significantly higher platelet count than the AGP and NAGP groups. Tao et al. [40] reported the blood platelet counts of suckling pigs at the age of one, four, and seven days as approximately 586.25 ± 115.42, 304.25 ± 123.00, and 406.75 ± 119.59 × 10^9^ cells per liter, respectively. Other factors that alter platelet count include animal health, age, stress, and internal bleeding [40]. The elevated platelet count in the HKL1 and HKL2 groups in this study suggests that HK L-137 may have some potential properties that modulate platelet creations.

It is well known that intestinal morphology is an essential index for gut health analysis. Increasing length and size of the intestinal villi induce absorptive ability by providing a greater surface area. Sayan et al. [41] reported that supplementation with probiotics improves the villus height, although there were no effects on suckling pigs’ crypt depth and villus height/crypt depth ratio. Supplementation with probiotics improved the villus height of the small intestine, thus enhancing the absorption capacity of the small intestine. *Lactobacillus* spp. can produce short-chain fatty acids to stimulate epithelial cells, enterocytes, mucus secretion, and the villus height and to promote the growth of the intestinal microflora [42,43,44]. However, there were no significant differences in the villus height, crypt depth, villus height/crypt depth ratio, and villus area among the treatment groups in the suckling and weaning pigs. This might be explained by the fact that HK l-137 is a paraprobiotic consisting of dead microbial cells, which cannot produce short-chain fatty acids to stimulate the morphological development of the intestinal epithelium in swine.

Weaning pigs can confront social, physical, and physiological stresses, increasing their intestinal absorption and depressing their immune function [39]. Previous studies demonstrated that HK L-137 could activate intestinal function and immune response [8,17,21,22,25,26]. The immunomodulating activity of paraprobiotics or non-viable microbes might result in some microbial components, for instance, exopolysaccharides (EPS), lipoteichoic acids (LTA), and other components, with anti-pathogen abilities [21,22,24,39]. Lipoteichoic acids (LTA) are cell membrane-bound polyglycerophosphate polymers anchored by a glycolipid and are highly substituted by D-alanyl esters [44]. Several studies reported that LTA is an effective activator or suppressor of the innate immune response via several mechanisms [35,45,46,47]. *Lactobacillus* strains can also produce EPS, which shields the surface polymer to prevent pathogens from attaching to intestinal barrier and regulate the release of pro-inflammatory cytokines [39,48].

Cytokines are a group of small proteins that specifically affect cell communications and are crucial in the regulation of the immune and inflammatory responses [45]. HK L-137 has been shown to trigger a significant increase in the production of several cytokines in human and animal models [21,22,26,27]. It has been shown to enhance beta (IFN-β) and gamma interferon (IFN-γ) production, promote NK-cell-killing activity, and activate macrophages, which protect against viral infection. To our best knowledge, the present study is the first to investigate the influence of HK L-137 on the expression of cytokine-encoding genes in suckling and starter pigs. A higher mRNA expression level of *IFN-β* was observed in the spleens of the suckling pigs (24-days-old) in the HK L-137 group, compared to the AGP and NAGP groups. These results are in agreement with the findings of Arimori et al. [26]. They reported that the daily intake of HK L-137 in pigs resulted in significantly higher mRNA expression levels of *IFN-β* in the whole blood cells than in the control group, which could enhance the host defense against influenza virus infection. Wang et al. [48] showed that a supplementation of *L. reuteri* I5007 could enhance T-cell differentiation and induce the mRNA expression of the ileal cytokine gene, indicating that *L. reuteri* I5007 strain can improve immune function in weaned piglets. The results are in line with Yu et al. [38], who demonstrated that probiotic supplementation increased serum-specific anti-OVA IgG levels. Moreover, *L. reuteri* has been observed to decrease proinflammatory cytokine mRNA expression, such as *IL-1β,* in the pig ileum [49].

A study by Hatano et al. [50] showed that HK L-137 was a more significant inducer of interleukin-12p40 (IL-12p40), a proinflammatory cytokine in the spleen cells of mice, which, if produced appropriately, will promote the immune response against the influenza virus. It was also demonstrated that this expression, which leads to the induction of *IL-12* in mouse spleen and splenic dendritic cells, may be attributed to the putative genes related to LTA synthesis in an internal plasmid of *L. plantarum L-137*. However, no significant difference in *IL-12* gene expression was found in this study. The higher levels of cytokine mRNA expression in the starter pigs in the HKL2 group compared to the AGP group might be attributed to antibiotic influence, which eliminated or disrupted the gut microbiota in the pigs. The cytokine system and its reaction are a complex trait resulting from an animal’s genotype and environment. To better understand the consequences of paraprobiotics on the interaction among the gut microbiome, the immune systems, and nutrient metabolism in swine and other livestock, a shotgun metagenomic analysis using high-throughput sequencing (HTS) should be performed.

## 5. Conclusions

A dietary supplementation at 20 mg/kg during the suckling phase to the weaning phase and at 4 mg/kg during the starter to the finisher phases increased growth performance and upregulated the expression of *IFN-β* and *IFN-γ* in pigs. It did not affect the villus height to crypt depth ratio. This study shows that the use of HK L-137 supplementation in suckling-to-fattening pigs might stimulate the immune response and promote growth performance in the finisher phase. Our study implies that HK L-137 supplementation may be a viable alternative to antibiotic growth promoters (AGPs) in the swine production industry.

## Figures and Tables

**Figure 1 vetsci-10-00087-f001:**
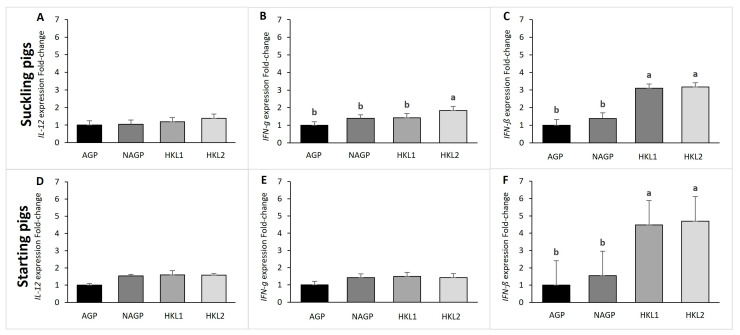
Effects of the dietary treatments on relative gene expression in the pigs at the end of the suckling phase (24 days old; (**A**) = *IL-12*, (**B**) = *IFN-γ*, (**C**) = *IFN-ß*) and the starter phase (73 days old; (**D**) = *IL-12*, (**E**) = *IFN-γ*, (**F**) = *IFN-ß*). AGP = basal feeds supplemented with antibiotics as a growth promoter (amoxicillin and colistin) at 300 ppm from the suckling to the grower phase; NAGP = basal feeds without AGP or any feed additive from the suckling to the finisher phases; HKL1 = basal feeds supplemented with HK L-137 at 20, 20, 20, 0, 0, and 0 ppm; and HKL2 = basal feeds supplemented with HK L-137 at 20, 20, 4, 4, 4, and 0 ppm during the suckling phase, weaner phase, starter phase, grower phase, finisher 1 phase, and finisher 2 phase, respectively. ^ab^ Means with different superscripts differ significantly (*p* < 0.05). SEM is the standard error mean. *IL-12 = Interleukin-12*, *IFN-ß = Interferon-β*, *IFN-γ = Interferon-γ*, and *ß-actin = Beta actin (control gene)*.

**Table 1 vetsci-10-00087-t001:** Composition and calculated nutrient content of the experimental diets from the suckling to the finisher 2 phases (g/100 g as the feed basis).

Item ^1^	Suckling	Weaner	Starter	Grower	Finisher 1	Finisher 2
	(14 Day–Weaned)	(Weaned–10 kg)	(10–25 kg)	(25–50 kg)	(50–75 kg)	(75–100 kg)
Ingredients						
Broken rice	50.40	45.85	46.00	35.00	38.00	54.00
Full fat soybean	18.00	30.00	30.00	20.00	-	-
Soybeans meal (44%)	3.00	10.00	14.00	18.00	26.00	16.00
Skim milk	13.00	5.00	1.00	-	-	-
Cornmeal	-	-	-	14.00	10.00	-
Defatted rice bran	-	5.00	5.00	-	-	-
Rice bran	-	-	-	10.00	20.00	25.00
Palm oil	-	-	-	-	3.00	2.00
Fish meals 60%	4.00	-	-	-	-	-
Fermented soybean meal	8.00	-	-	-	-	-
Mono-dicalcium phosphate	0.10	-	-	-	-	-
Dicalcium phosphate	-	2.00	2.00	1.80	1.30	1.30
Bone meals	2.00	-	-	-	-	-
Calcium carbonate	0.10	0.60	0.50	0.20	0.70	0.60
Salt	0.40	0.50	0.50	0.50	0.50	0.50
L-lysine	0.30	0.40	0.30	-	-	0.10
DL-methionine	0.30	0.20	0.10	-	-	-
Threonine	0.15	0.20	0.10	-	-	-
Premix	0.25	0.25	0.50	0.50	0.50	0.50
Chemical composition						
Crude Protein, %	22.32	20.09	21.24	20.15	18.00	14.76
ME, kcal/kg	3346	3341	3396	3315	3316	3290
Crude Fiber, %	1.43	2.79	3.06	3.68	3.97	3.59
Ether Extract, %	4.48	6.93	6.91	5.37	1.44	1.19
Lysine, %	1.71	1.62	1.53	1.14	1.00	0.86
Methionine, %	0.75	0.66	0.45	0.33	0.31	0.27
Threonine, %	1.08	1.02	0.93	0.79	0.69	0.56
Tryptophan, %	0.29	0.29	0.29	0.27	0.24	0.20
Calcium, %	1.05	0.90	0.83	0.64	0.68	0.62
Available Phosphorus, %	0.61	0.44	0.40	0.37	0.30	0.29

^1^ The dietary treatments were divided into four treatments as follows: (1) AGP = basal feeds supplemented with antibiotics as a growth promoter (amoxicillin and colistin) at 300 mg/kg from the suckling to grower phases; (2) NAGP = basal feeds without AGP or any feed additive from the suckling to the finisher phases; (3) HKL1 = basal feeds supplemented with HK L-137 at 20, 20, 20, 0, 0, and 0 mg/kg; and (4) HKL2 = basal feeds supplemented with HK L-137 at 20, 20, 4, 4, 4, and 0 mg/kg during the suckling phase, weaner phase, starter phase, grower phase, finisher 1 phase, and finisher 2 phase, respectively.

**Table 2 vetsci-10-00087-t002:** Primer sequences for the semi-quantitative real-time PCR assay.

Gene *	Primer Sequences (5′–3′)	PCR Size (bp)	Accession No.
*IL-12*	F: AAG CTG TTC ACA AGC TCA AGT ATG A	81	NM_214013.1
	R: TCT TGG GAG GGT CTG GTT TG		
*IFN-ß*	F: TGC AAC CAC CAC AAT TCC	79	NM_001003923.1
	R: CTG AGA ATG CCG AAG ATC TG		
*IFN-γ*	F: TGG TAG CTC TGG GAA ACT GAA TG	79	NM_213948.1
	R: GGC TTT GCG CTG GAT CTG		
*ß-actin*	F: CTC CTT CCT GGG CAT GGA	65	U07786.1
	R: CGC ACT TCA TGA TCG AGT TGA		

* IL-12 = Interleukin-12, IFN-ß = Interferon-β, IFN-γ = Interferon-γ, and ß-actin = Beta actin (control gene).

**Table 3 vetsci-10-00087-t003:** Effects of the dietary treatments on the growth performances of piglets fed on the experimental diets from 14 days of age until weaned (24 days old).

Parameters	Dietary Treatments				SEM	*p*-Value
	AGP	NAGP	HKL1	HKL2		
Number of suckling pigs	118	133	117	125		
Litter size						
Total born	12.22	14.3	12.3	13.90	0.96	0.385
Born alive	11.89	13.3	11.7	12.50	0.89	0.642
Piglet body weight						
At birth, kg	1.53	1.48	1.60	1.58	0.06	0.570
At 14 days old, kg	4.64	4.26	4.44	4.67	0.17	0.192
At 24 days old, kg	5.70	5.48	5.67	5.81	0.20	0.712
Weight gain, kg						
At birth–d14	3.11	2.78	2.84	3.09	0.18	0.241
At 14–24 days old	1.08	1.22	1.23	1.14	0.19	0.716
At birth–24 days old	4.19	3.99	4.07	4.23	0.21	0.853
Average daily gain, g/d						
At birth–14 days old	222	198	203	221	12.70	0.241
At 14–24 days old	108	122	123	114	19.28	0.716
At birth–24 days old	182	174	183	184	9.20	0.853
Average daily feed intake, g/d	205	189	209	204	19.99	0.912
Feed efficiency (Gain: Feed)	0.53	0.64	0.59	0.56	0.09	0.881
Diarrhea incidence, %						
Average at birth–14 days old	11.83	10.97	12	11.49	1.12	0.924
Average at 14–24 days old	6.00	6.96	7.68	7.68	0.90	0.528

AGP = basal feeds supplemented with antibiotics as a growth promoter (amoxicillin and colistin) at 300 ppm from the suckling phase to the grower phase; NAGP = basal feeds without AGP or any feed additive from the suckling to finisher phases; HKL1 = basal feeds supplemented with HK L-137 at 20, 20, 20, 0, 0 and 0 ppm; and HKL2 = basal feeds supplemented with HK L-137 at 20, 20, 4, 4, 4, and 0 ppm during the suckling phase, weaner phase, starter phase, grower phase, finisher 1 phase 1, and finisher 2 phase, respectively. SEM is the standard error mean.

**Table 4 vetsci-10-00087-t004:** Effects of the dietary treatments on the growth performances of the pigs from the weaner phase to the starter phase (24 to 66 days old).

Parameters	Dietary Treatments				SEM	*p*-Value
	AGP	NAGP	HKL1	HKL2		
Initial weight (24 days old)	6.10	5.94	5.93	6.13	0.07	0.699
Body Weight, kg						
38 days	8.11	7.60	7.64	7.71	0.10	0.232
52 days	14.45	13.07	13.46	13.80	0.20	0.079
66 days	21.42	19.90	20.14	20.42	0.26	0.164
Weight gain, kg						
24–38 days	2.02	1.66	1.72	1.58	0.07	0.157
38–52 days	6.34 ^a^	5.47 ^b^	5.82 ^ab^	6.09 ^a^	0.12	0.034
52–66 days	6.97	6.83	6.67	6.63	0.10	0.652
Overall	15.32	13.97	14.21	14.30	0.23	0.153
Average daily gain, g/d						
24–38 days	144	119	123	113	5.17	0.157
38–52 days	453 ^a^	391 ^b^	416 ^ab^	435 ^a^	8.22	0.034
52–66 days	498	488	477	473	7.27	0.652
Overall	365	333	338	340	5.42	0.153
Average daily feed intake, g/d						
24–38 days	239	203	207	199	6.18	0.081
38–52 days	745	662	645	709	18.57	0.214
52v66 days	908	915	885	891	13.27	0.862
Overall	631	593	579	600	9.00	0.225
Feed efficiency (Gain: Feed)						
24–38 days	0.60	0.58	0.54	0.56	0.02	0.839
38–52 days	0.62	0.59	0.64	0.62	0.01	0.619
52v66 days	0.55	0.53	0.54	0.53	0.01	0.755
Overall	0.58	0.56	0.58	0.57	0.01	0.532

AGP = basal feeds supplemented with antibiotics as a growth promoter (amoxicillin and colistin) at 300 ppm from the suckling phase to the grower phase; NAGP = basal feeds without AGP or any feed additive from the suckling to the finisher phases; HKL1 = basal feeds supplemented with HK L-137 at 20, 20, 20, 0, 0, and 0 ppm; and HKL2 = basal feeds supplemented with HK L-137 at 20, 20, 4, 4, 4, and 0 ppm during the suckling phase, weaner phase, starter phase, grower phase, finisher 1 phase, and finisher 2 phase, respectively. ^ab^ Means with different superscripts differ significantly. SEM is the standard error mean.

**Table 5 vetsci-10-00087-t005:** Effects of the dietary treatments on the growth performances of the pigs from the starter to the finisher phases to reach 100 kg in body weight.

Parameters	Dietary Treatments				SEM	*p*-Value
	AGP	NAGP	HKL1	HKL2		
Grower phase, 25–50 kg						
ADG, kg/d	0.67	0.54	0.60	0.58	0.05	0.450
ADFI, kg/d	1.66	1.43	1.49	1.42	0.12	0.645
Feed efficiency (Gain: Feed)	0.41	0.38	0.41	0.41	0.01	0.179
Experimental periods	37 ^a^	46 ^c^	42 ^ab^	43 ^bc^	1.56	0.002
Finisher phase 1, 50–75 kg						
ADG, kg/d	0.90	0.79	0.92	0.92	0.06	0.418
ADFI, kg/d	2.46	2.20	2.52	2.50	0.17	0.530
Feed efficiency (Gain: Feed)	0.36	0.36	0.36	0.37	0.10	0.596
Experimental periods	28	32	27	27	1.53	0.063
Finisher phase 2, 75–100 kg						
ADG, kg/d	0.98	1.00	0.95	1.10	0.03	0.065
ADFI, kg/d	2.82	2.99	2.79	3.05	0.09	0.273
Feed efficiency (Gain: Feed)	0.35 ^ab^	0.34 ^b^	0.35 ^ab^	0.36 ^a^	0.02	0.017
Experimental periods	26	25	26	23	1.10	0.194
Grower–Finisher phase 2, 25–100 kg						
ADG, kg/d	0.82 ^a^	0.73 ^b^	0.79 ^a^	0.81 ^a^	0.06	0.015
ADFI, kg/d	2.31	2.21	2.27	2.32	0.06	0.512
Feed efficiency (Gain: Feed)	0.36 ^a^	0.33 ^b^	0.35 ^a^	0.35 ^a^	0.01	0.018
Experimental periods	91 ^a^	103 ^b^	95 ^a^	93 ^a^	1.91	<0.001

AGP = basal feeds supplemented with antibiotics as a growth promoter (amoxicillin and colistin) at 300 ppm from the suckling phase to the grower phase; NAGP = basal feeds without AGP or any feed additive from the suckling to finisher phases; HKL1 = basal feeds supplemented with HK L-137 at 20, 20, 20, 0, 0, and 0 ppm; and HKL2 = basal feeds supplemented with HK L-137 at 20, 20, 4, 4, 4, and 0 ppm during the suckling phase, weaner phase, starter phase, grower phase, finisher 1 phase, and finisher 2 phase, respectively. ^abc^ Means with different superscripts differ significantly (*p* < 0.05). SEM is the standard error mean.

**Table 6 vetsci-10-00087-t006:** Effects of the dietary treatments on the hematological parameters of the pigs at the end of the suckling phase (24 days old) and starter phase (73 days old).

Parameters	Dietary Treatments				SEM	*p*-Value
	AGP	NAGP	HKL1	HKL2		
Suckling phase (24 days old)						
RBC count, 10^6^/μL	7.16	7.39	7.61	7.70	0.33	0.371
Hb, g/dl	13.3	14.15	14.13	14.88	0.44	0.169
Hct, %	42.5	44.35	44.75	47.75	1.39	0.130
MCV, fl	58.50	60.75	58.50	60.75	1.76	0.737
MCH, fg	18.88	19.40	18.58	19.28	0.59	0.825
MCHC, g/dl	32.08	31.90	31.75	31.83	0.16	0.648
RDW, %	17.55	16.53	15.28	16.15	0.45	0.064
WBC count, cell/cu.mm.	21,700	17,825	19,475	13,800	2304	0.190
Neutrophil, %	42.75	45.75	44.75	39.75	3.23	0.602
Lymphocyte, %	56.50	53.75	54.75	60.25	3.19	0.535
Monocyte, %	0.00	1.00	1.00	0.00	0.14	0.248
Platelet count, cell/cu.mm.	264,000 ^b^	218,250 ^b^	484,500 ^a^	575,750 ^a^	58,279	0.003
Starter phase (73 days old)						
RBC, 10^6^/µL	7.33	7.11	7.18	7.50	0.15	0.314
Hb, g/dl	13.18	12.94	13.30	12.94	0.28	0.787
Hct, %	41.60	40.80	41.60	41.00	0.87	0.905
MCV, fl	56.80	56.80	57.80	54.40	1.09	0.204
MCH, fg	18.02	18.18	18.5	17.28	0.35	0.139
MCHC, g/dl	31.70	31.86	31.98	31.68	0.16	0.641
RDW, %	14.30	14.22	14.06	14.40	0.29	0.877
WBC, cell/cu.mm.	16,640	19,000	16,420	20,040	1958	0.536
Neutrophil, %	26.80	31.80	27.40	25.40	5.12	0.864
Lymphocyte, %	70.80	67.00	71.40	71.60	5.21	0.933
Monocyte, %	1.00	1.25	1.00	2.67	0.41	0.413
Eosinophil, %	4.50	1.00	1.00	1.60	0.50	0.386
Platelet count, cell/cu.mm.	259,200	233,400	281,400	249,000	27,492	0.706

AGP = basal feeds supplemented with antibiotics as a growth promoter (amoxicillin and colistin) at 300 ppm from the suckling phase to the grower phase; NAGP = basal feeds without AGP or any feed additive from the suckling to the finisher phases; HKL1 = basal feeds supplemented with HK L-137 at 20, 20, 20, 0, 0, and 0 ppm; and HKL2 = basal feeds supplemented with HK L-137 at 20, 20, 4, 4, 4, and 0 ppm during the suckling phase, weaner phase, starter phase, grower phase, finisher 1 phase, and finisher 2 phase, respectively. ^ab^ Means with different superscripts differ significantly (*p* < 0.05). SEM is the standard error mean.

**Table 7 vetsci-10-00087-t007:** Effects of the dietary treatments on the small intestine (duodenum, jejunum, and ileum) histology of suckling pigs (24 days old).

Items	Dietary Treatments				SEM	*p*-Value
	AGP	NAGP	HKL1	HKL2		
Suckling phase (24 days old)						
Duodenum						
Villus height, µm	300.13	278.27	275.57	347.00	28.96	0.363
Crypt depth, µm	118.50	112.14	101.37	137.37	10.69	0.197
Villus height/crypt depth ratio	2.67	2.48	2.94	2.54	0.40	0.885
Villus area, mm^2^	20.43	21.38	19.91	28.64	2.65	0.178
Jejunum						
Villus height, µm	293.82	270.91	262.52	227.71	32.62	0.610
Crypt depth, µm	121.88	108.92	106.99	109.91	12.82	0.876
Villus height/crypt depth ratio	2.45	2.56	2.45	2.07	0.26	0.602
Villus area, mm^2^	21.93	18.45	20.28	14.88	3.56	0.629
Ileum						
Villus height, µm	210.61	159.16	257.26	198.46	25.79	0.120
Crypt depth, µm	111.29	88.57	120.83	96.55	11.51	0.263
Villus height/crypt depth ratio	2.00	1.78	2.16	2.22	0.33	0.827
Villus area, mm^2^	13.07	10.72	21.35	11.24	3.09	0.222
Starter phase (73 days old)						
Duodenum						
Villus height, µm	265.52	286.05	272.43	250.12	18.79	0.620
Crypt depth, µm	136.25	130.52	125.55	127.55	9.77	0.882
Villus height/crypt depth ratio	1.96	2.20	2.20	2.00	0.14	0.510
Villus area, mm^2^	21.97	22.94	27.30	22.80	3.21	0.740
Jejunum						
Villus height, µm	322.31	317.52	318.96	289.37	19.18	0.622
Crypt depth, µm	125.76	127.56	125.55	133.87	11.89	0.958
Villus height/crypt depth ratio	2.62	2.61	2.69	2.19	0.28	0.598
Villus area, mm^2^	27.13	24.24	27.10	22.95	1.54	0.228
Ileum						
Villus height, µm	223.02	249.66	245.43	228.44	20.53	0.765
Crypt depth, µm	108.62	118.15	113.67	112.98	10.70	0.940
Villus height/crypt depth ratio	2.11	2.12	2.20	2.06	0.17	0.959
Villus area, mm^2^	16.40	19.00	16.93	16.28	2.1	0.784

AGP = basal feeds supplemented with antibiotics as a growth promoter (amoxicillin and colistin) at 300 ppm from the suckling phase to the grower phase; NAGP = basal feeds without AGP or any feed additive from the suckling to the finisher phases; HKL1 = basal feeds supplemented with HK L-137 at 20, 20, 20, 0, 0, and 0 ppm; and HKL2 = basal feeds supplemented with HK L-137 at 20, 20, 4, 4, 4, and 0 ppm during the suckling phase, weaner phase, starter phase, grower phase, finisher 1 phase, and finisher 2 phase, respectively.

## Data Availability

The data that support the findings of this study are available from the corresponding author upon reasonable request.

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
