# Peer review of "Effects of Heat-Killed Lactobacillus plantarum L-137 Supplementation on Growth Performance, Blood Profiles, Intestinal Morphology, and Immune Gene Expression in Pigs"

_vetsci, 2023, doi:10.3390/vetsci10020087_

Round 1
Reviewer 1 Report
1. Introduction: please give more detail about metabolites from Lactobacillus plantarum strain L-137 (not only the effects of Lactobacillus plantarum strain L-137)
2. Lines: 298-291 have to be deleted
3. Line 381: Are lipoteichoic acids (LTA) also produced from Lactobacillus plantarum strain L-137?
3. In discussion: The metabolites (type) from Lactobacillus plantarum strain L-137 should be indicated and referred to explain the results (์not only LTA)
Author Response
Thank you so much for your kind suggestions. I have already responded to your question as follows:
- Introduction: please give more detail about metabolites from Lactobacillus plantarum strain L-137 (not only the effects of Lactobacillus plantarum strain L-137)
- We have already revised
- Lines: 298-291 have to be deleted
- We have already revised
- Line 381: Are lipoteichoic acids (LTA) also produced from Lactobacillus plantarum strain L-137?
- Yes, Lipoteichoic acids (LTA) are found in the cell walls of Gram-positive bacteria.
- In discussion: The metabolites (type) from Lactobacillus plantarum strain L-137 should be indicated and referred to explain the results (์not only LTA)
- We have already revised in discussion.
Reviewer 2 Report
The current manuscript described the effects of heat-killed Lactobacillus on performance and measures of blood profiles, gut morphology, and gene expression related to inflammation in pigs. The current manuscript must have extensive revision for further process with critical corrections regarding logical, grammatical, and editorial errors.
1. Firstly, there is no substantial justification for why this study needs to look into the effects of HK-L137 during the whole growth phases from suckling to Finisher. According to the descriptions in the introduction part, the nursery phase (early weaner + Starter) would be enough to answer the questions in the introduction without other phases. If the authors do not have professional reasons for using whole growth phases in this study, you may need to change the data size using only the nursery.
2. In the introduction part, there is a need to briefly describe information about the microbes used for this study. At least, why authors test this microbe, why we need to have heat treatment for the microbe rather than using an intact source, and possible mechanisms of this based on previous findings. In the introduction, you should describe what you have found for heat-treated probiotics in animals and then what you did not know.
3. Again, the authors should clarify the hypothesis. Need more explanation about the background of why authors have to measure blood profiles, gut morphology in each place (duodenum, jejunum, and ileum), and immune gene expression in the spleen (why spleen). Thus, this leads to setting a hypothesis (‘little information’ cannot be a hypothesis). However, the current introduction makes me unclear what the importance or uniqueness of this study as compared to previous experiments. Please add more information regarding the importance and uniqueness of this study in the introduction.
4. An experimental design did not appear in the M&M part.
5. For early weaner and starters, these diets contained relatively low levels of animal protein sources compared with others. Also, this study used the experimental diets containing broken rice and fully fat soybean, which could not be the feedstuff commonly used for swine feeds rather than corn and soybean meal. Readers could be wondering about the specs of the feeds if they were surely meeting the NRC requirements. So please additionally describe the calculated composition of the feeds (ME value, SID Lys, Met, Thr, Trp, Ca, available P).
6. In discussion, you should provide more professional reasons considering various aspects (biological or biochemical knowledges) why HKL probiotics did not cause difference on the measurements in this study. Authors described their speculations based on the results at the end of each paragraphs, but authors may need to describe professional and scientific reasons based on previous findings. The most statements in the discussion are saying like this trial was failed to detect the dietary effects due to various factors. Why HKL lead to the growth change based on its potential mechanisms based on the measures. You a bit better described the mechanism regarding impacts of HKL (using specialty having difference from other probiotics) based on some biological knowledge.
Author Response
Thank you so much for your kind suggestions. I have already responded to your question as follows:
- Firstly, there is no substantial justification for why this study needs to look into the effects of HK-L137 during the whole growth phases from suckling to Finisher. According to the descriptions in the introduction part, the nursery phase (early weaner + Starter) would be enough to answer the questions in the introduction without other phases. If the authors do not have professional reasons for using whole growth phases in this study, you may need to change the data size using only the nursery.
- The weaning of piglets is acutely stressful during the initial stage of life, causing a reduction in feed intake, increased incidence of diarrhea, growth retardation, susceptibility to diseases, and increased mortality. This influence will have an effect on pig performance throughout the finishing stage.
- HK L-137 might activate the intestinal function, modulate the immune response, and induce an augmentation of productive performance, the supplementation period, and its effect on immune-related gene expression in suckling to fattening pigs have not been studied in detail.
- Our study indicates that supplementing HK L-137 in pig diets can improve production performance similar to using AGP in finishing pigs
- In the introduction part, there is a need to briefly describe information about the microbes used for this study. At least, why authors test this microbe, why we need to have heat treatment for the microbe rather than using an intact source, and possible mechanisms of this based on previous findings. In the introduction, you should describe what you have found for heat-treated probiotics in animals and then what you did not know.
- Due to variations in preparation techniques, feed storage, and their capacity to survive transit through the stomach and multiply in the intestine, the effects of probiotic supplementation may not be uniform. Additionally, there may be some issues with the animal production business due to the growing usage of probiotics that could release live bacteria into the environment.
- Our previous studies reported that non-viable microbes referred to as immunobiotics or paraprobiotics, also exhibit beneficial effects on the immune response equivalent to or greater than probiotics. Moreover, they still have no limitations on the processing, storage, and ability to survive in the gastrointestinal tracts.
- HK L-137 may activate the intestinal function, regulate immunological response, and cause an increase in productive performance during the supplementation time. However, its impact on immune-related gene expression in suckling to fattening pigs has yet to be thoroughly researched.
- Again, the authors should clarify the hypothesis. Need more explanation about the background of why authors have to measure blood profiles, gut morphology in each place (duodenum, jejunum, and ileum), and immune gene expression in the spleen (why spleen). Thus, this leads to setting a hypothesis (‘little information’ cannot be a hypothesis). However, the current introduction makes me unclear what the importance or uniqueness of this study as compared to previous experiments. Please add more information regarding the importance and uniqueness of this study in the introduction.
- Probiotic bacteria have been used as an alternative to antibiotics in pig production. Nevertheless, in-feed probiotics may be inconsistent due to differences in preparation methods, feed storage, and their ability to survive through the gastrointestinal tract.
- Heat-killed Lactobacillus plantarum L-137 (HK L-137), also known as paraprobiotics, has been shown to have beneficial effects on the immune response and has no limitations. Our study indicates that supplementing HK L-137 in pig diets can improve production performance similar to using AGP in finishing pigs and promote immune function.
- An experimental design did not appear in the M&M part.
- We have revised experimental design in the M&M part.
- For early weaner and starters, these diets contained relatively low levels of animal protein sources compared with others. Also, this study used the experimental diets containing broken rice and fully fat soybean, which could not be the feedstuff commonly used for swine feeds rather than corn and soybean meal. Readers could be wondering about the specs of the feeds if they were surely meeting the NRC requirements. So please additionally describe the calculated composition of the feeds (ME value, SID Lys, Met, Thr, Trp, Ca, available P).
- We were unable to offer in this section.
- In discussion, you should provide more professional reasons considering various aspects (biological or biochemical knowledges) why HKL probiotics did not cause difference on the measurements in this study. Authors described their speculations based on the results at the end of each paragraphs, but authors may need to describe professional and scientific reasons based on previous findings. The most statements in the discussion are saying like this trial was failed to detect the dietary effects due to various factors. Why HKL lead to the growth change based on its potential mechanisms based on the measures. You a bit better described the mechanism regarding impacts of HKL (using specialty having difference from other probiotics) based on some biological knowledge.
- Thank you so much for your suggestion; we have revised it to include more discussion.
Reviewer 3 Report
The manuscript overall is well written. The heat killed L. plantarum was previously widely investigated and there are quite much of information about it's positive effect. An important part within the manuscript is associated with testing of the expression of cytokine-encoding genes. The attention for possible use alternatives instead of antibiotics also is very important.
For further consideration of the manuscript I would suggest to pay attention to these points:
- the most important point which makes me feel skeptical in obtained results is associated with the dosage of HK L-137:
you prove that the product contains 1 × 1011 cells/g Lactobacillus plantarum in the dry product. The dosage of the product was 20mg/kg (or even 4 mg/kg). Firstly, it is unclear do kg means live weight of the animal or kg of feed? Please add this information. The second point is that if to give to piglet 20 mg/kg of the product it will contain 20,22 cells/kg. If the weight of piglet is 10 kg it will obtain 200 cells of L. plantarum. It is very low amount in comparing with alive of even dead bacteria (including Lactobacillus spp) in live organism and I did not believe that such a small amount can do influence to immunity or other physiological parameters. Therefore, I suggest to check whether it was really such small dosage of the product given to piglets?
Minor comments
-sentences in lines 289-292 should be dropped out from the text;
-sentence in lines 311-313 about the increase of platelets should be re-arranged, trying to describe possible mechanism or reason why the increasing can occur in this case rather only to state that "HK L-137 allows animals to create platelets to prevent injury to body cells or internal bleeding".
-line 368: what is the HK L-137 group? There was no such group in experiments, probably it should be HKL1, or HKL2 or groups supplemented with HK L-137? Please check.
-in abstract (lines 32-36) you have mentioned 4 groups of piglets but the group HKL1 is not mentioned as HKL1 (3d group), Please add those letters (HKL1) to mention all of the groups (AGP, NAGP, HKL1 and HKL2) similarly.
- data availability statement (lines 414-415) should be corrected (last 5 words should be deleted);
References style are not in accordance to MDPI journal style. Please check the latest articles from Vet Sci journal to check the style of the references.
Author Response
Thank you so much for your kind suggestions. I have already responded to your question as follows:
you prove that the product contains 1 × 1011 cells/g Lactobacillus plantarum in the dry product. The dosage of the product was 20mg/kg (or even 4 mg/kg). Firstly, it is unclear do kg means live weight of the animal or kg of feed? Please add this information. The second point is that if to give to piglet 20 mg/kg of the product it will contain 20,22 cells/kg. If the weight of piglet is 10 kg it will obtain 200 cells of L. plantarum. It is very low amount in comparing with alive of even dead bacteria (including Lactobacillus spp) in live organism and I did not believe that such a small amount can do influence to immunity or other physiological parameters. Therefore, I suggest to check whether it was really such small dosage of the product given to piglets?
- We have revised to 1 × 1011 cfu/g Lactobacillus plantarum in the dry product, and used by mg per kg dietary treatments
Minor comments
-sentences in lines 289-292 should be dropped out from the text;
- We have already deleted it.
-sentence in lines 311-313 about the increase of platelets should be re-arranged, trying to describe possible mechanism or reason why the increasing can occur in this case rather only to state that "HK L-137 allows animals to create platelets to prevent injury to body cells or internal bleeding".
- We have already revised it.
-line 368: what is the HK L-137 group? There was no such group in experiments, probably it should be HKL1, or HKL2 or groups supplemented with HK L-137? Please check.
- We have already revised it.
-in abstract (lines 32-36) you have mentioned 4 groups of piglets but the group HKL1 is not mentioned as HKL1 (3d group), Please add those letters (HKL1) to mention all of the groups (AGP, NAGP, HKL1 and HKL2) similarly.
- We have already revised it.
- data availability statement (lines 414-415) should be corrected (last 5 words should be deleted);
- We have already revised it.
References style are not in accordance to MDPI journal style. Please check the latest articles from Vet Sci journal to check the style of the references.
- We have already revised it.
Reviewer 4 Report
In the present study, the authors investigated the effects of HK bacteria on the productive performance, intestinal morphology, and cytokine gene expression of suckling to fattening pigs. Specific comments as follow:
1. Does the addition of HKL-137 increase the gross energy of feeds?
2. Does the 90% of whey protein, dextrin, and sunflower lecithin in HKL-137 influence the appetite of piglets? Leptin level or some hormones related to appetite may be suggested to test, if the serum was collected in fast status.
3. If the appetite has been influenced, the administration of lactobacillus plantarum by feed may be not proper.
4. The experiment design is confusing in line 105-109. How did the authors select piglets?
5. It should be better to draw a diagram for experiment design in this study.
6. Why did the authors choose 14 days suckling piglets as start? Can the author confirm the volume of milk intake from sows for every piglet? If not, how could the author calculate the Feed efficiency?
7. Only one p value is not enough and strict for the data. Antibiotics and HK are two factors. It is suggested to re-analyze the data.
8. Please provide the number of piglets in table 3.
9. Culling piglets is important for growth performance analysis. Please add the information for culling rate of test piglets from starter to finish.
10. Please provide the refence range for every index.
11. Compared with AGP, increase of IFN in NAGP, but also increase in HK? Confused.
12. Line 102, 1x1011 cells, revise it.
13. Line 160, the method here is real time quantitative RT-PCR or semiquantitative RT-PCR? Confused.
Author Response
Thank you so much for your kind suggestions. I have already responded to your question as follows....
- Does the addition of HKL-137 increase the gross energy of feeds?
- HK L-137 has no effect on the gross energy of feed.
- Does the 90% of whey protein, dextrin, and sunflower lecithin in HKL-137 influence the appetite of piglets? Leptin level or some hormones related to appetite may be suggested to test, if the serum was collected in fast status.
- Thank you for your suggestions; however, we did not test whether the ingredient of HK L137 was related to piglet appetite or not. In the upcoming study will look at leptin levels.
- If the appetite has been influenced, the administration oflactobacillus plantarum by feed may be not proper.
- We believed that HK L-137, which was added in small amounts as imunobiotics to improve immune response, had no impact on the piglet's appetite.
- The experiment design is confusing in line 105-109. How did the authors select piglets?
- We have already revised, The crossbred [(Large White × Landrace) × Duroc; 4.5 ±0.54 kg initial body weight (BW)] suckling pigs were used and assigned to each of 4 dietary treatments based on piglets from 10 a total of 40 healthy sows (Landrace × Large White, 250±50 kg of BW) at parity 2-6 which gave birth to full term and had the same litter size (more than ten live piglets per litter), sows per treatment. Experimental feeds (Creep feed) were offered from 14 days (suckling pigs) to 24 days at weaning age.
- It should be better to draw a diagram for experiment design in this study.
- We have already revised the experimental design.
- Why did the authors choose 14 days suckling piglets as start? Can the author confirm the volume of milk intake from sows for every piglet? If not, how could the author calculate the Feed efficiency?
- We are curious to know the impact of HK L-137 on the immune responses of 14 day piglets because this is a critical time. The quantity of milk consumed was not measured. We just concentrated on our creep feed experiment.
- Only one p value is not enough and strict for the data. Antibiotics and HK are two factors. It is suggested to re-analyze the data.
- Thank you for your suggestion, we have already re-analyzed.
- Please provide the number of piglets in table 3.
- We have already added the number of piglets in Table 3
- Culling piglets is important for growth performance analysis. Please add the information for culling rate of test piglets from starter to finish.
- There were no culling pig in this experiment.
- Please provide the refence range for every index.
- We cannot provide the refence range for every blood parameters of pigs, we use the average of control group as references.
- Compared with AGP, increase of IFN in NAGP, but also increase in HK? Confused.
- There were no significant difference between AGP and NAGP in gene expression analysis
- Line 102, 1x1011 cells, revise it
- Already revised
- Line 160, the method here is real time quantitative RT-PCR or semiquantitative RT-PCR? Confused.
- Already revised
Round 2
Reviewer 2 Report
There are some questionable points in some data tables. Once, if a fundamental error occurs again, it will be difficult to proceed with the further process for future publication. Authors, please keep this in mind and proceed with further revisions.
5 Author name is different from the list in citation info. Please double-check.
39, greater ADG
41 should indicate the location ‘ a higher platelet count in xxxx’
42 In the Abstract, the results could not support the conclusion. The conclusion sentence in abstract should be reworded. 1) should specify the period of the improved growth. 2) Also the immune response was not promoted by treatment.
87-88 grammar error: not a complete sentence
101 please briefly describe how to prepare the product, with citation
104 ‘cfu/g of’
109 change ‘healthy’ to ‘lactating’, healthy would be subjective in that sentence.
127 please delete the NRC, the calculated composition was not following NRC 2012 recommendation.
Table 3, Different to Table 4. Why weight gain and ADG have different P-values? I think it would be same. were there outlier? Please double check the analysis.
Table 3, At birth to 24 d old, the numbers of their diarrhea incidence seems sum of (At birth to 14 d old) and (A14 to 24 d old). Please double check the analysis.
203 please clearly separate the footnote and main contents.
208 change ‘better’ to ‘greater’
Table 4. 24 days old
Table 4. In feed efficiency, the data in Overall has smaller difference between treatment with same SEM compared with the data in 38 to 52. However, the p-value in overall is lower than 38 to 52 d. It is hard to accept for me. Please check the data and describe the each data with three digits. I am not an expert in statistics, but need explantation on this.
Table 5. Change ‘FE’ to ‘Feed efficiency (G:F)’
Table 5. Change ‘0.000’ to ‘<0.001’
Author Response
Thank you very much for your comments and suggestions.
5 Author name is different from the list in citation info. Please double-check.
- Already revised.
39, greater ADG
- Already revised.
41 should indicate the location ‘ a higher platelet count in xxxx’
- We have revised to a higher platelet count (484,500 and 575,750) than in others.
42 In the Abstract, the results could not support the conclusion. The conclusion sentence in abstract should be reworded. 1) should specify the period of the improved growth. 2) Also the immune response was not promoted by treatment.
- We have revised to ‘These present studies demonstrate that the HK L-137 supplementation could stimulate immune response in suckling and starting pig and promote the growth performance in finishing pigs’.
87-88 grammar error: not a complete sentence
- Already revised.
101 please briefly describe how to prepare the product, with citation
- Already revised.
104 ‘cfu/g of’
- Already revised.
109 change ‘healthy’ to ‘lactating’, healthy would be subjective in that sentence.
- Already revised.
127 please delete the NRC, the calculated composition was not following NRC 2012 recommendation.
- For early weaners and starters, our study used the experimental diets containing broken rice and fully fat soybean, which are commonly used for swine feeds in Thailand and have nutrient composition not lower than the NRC requirement. We have described the calculated composition of the feeds such as ME,Lys, Met, Thr, Trp, Ca, and available P in Table 3 and revised line129 to All nutrients of dietary treatment met or exceeded the NRC requirements.
Table 3, Different to Table 4. Why weight gain and ADG have different P-values? I think it would be same. were there outlier? Please double check the analysis.
- This is the error form the three-digit adjustment, we have already revised.
Table 3, At birth to 24 d old, the numbers of their diarrhea incidence seems sum of (At birth to 14 d old) and (A14 to 24 d old). Please double check the analysis.
- We have removed % diarrhea incidence at birth to 24 d.
203 please clearly separate the footnote and main contents.
- Already revised.
208 change ‘better’ to ‘greater’
- Already revised.
Table 4. 24 days old
- Already revised.
Table 4. In feed efficiency, the data in Overall has smaller difference between treatment with same SEM compared with the data in 38 to 52. However, the p-value in overall is lower than 38 to 52 d. It is hard to accept for me. Please check the data and describe the each data with three digits. I am not an expert in statistics, but need explantation on this.
- We would like to confirm that our data is correct. We double-checked and found the same that there was no significant difference between the groups.
Table 5. Change ‘FE’ to ‘Feed efficiency (G:F)’
- Already revised.
Table 5. Change ‘0.000’ to ‘<0.001’
- Already revised.
Reviewer 3 Report
The authors improved their manuscript according to my suggestions. However, before the submission of the last version of the manuscript the authors should pay attention to the following points:
1. In lines 104-105 you have changed the concentration of bacteria and the sentence now sounds as " It contains 1 × 1011 cfu/g Lactobacillus plantarum in the dry product stored at room temperature [21]". In the reference 21 I was unable to find this information, therefore you should one more time to check precisely an information what exact concentration was used of bacteria in the product as supplement and to provide it in the manuscript with appropriate reference.
2. I would suggest to remove or to modify the sentence in conclusions in lines 414-416: "This is the first study to show that HK L-137 supplementation from suckling to fattening pig might stimulate immune response and improve growth performance in the finisher stage in the same way that antibiotic growth promoters could". Firstly there were studies on immune response previously (for instance, doi: 10.3109/08923973.2012.672425); the growth performance and some immune changes in pigs also were previously investigated by authors (Watcharin Omthonglang et al., Effect of Cordyceps militaris spent mushroom substrate and Heat-killed Lactobacillus plantarum (HK L-137) supplementation in nursery pig diets on production efficiency, oxidative, and immune status). The words "in the same way" also may be incorrect if to compare action mechanism of antibiotics and HK L-137.
Author Response
- In lines 104-105 you have changed the concentration of bacteria and the sentence now sounds as " It contains 1 × 1011cfu/g Lactobacillus plantarum in the dry product stored at room temperature [21]". In the reference 21 I was unable to find this information, therefore you should one more time to check precisely an information what exact concentration was used of bacteria in the product as supplement and to provide it in the manuscript with appropriate reference.
- Thank you for your comments. According to the concentration from the company’s product instruction, we have deleted the reference [21] from the sentence and replaced it with (information from product instruction)
- I would suggest to remove or to modify the sentence in conclusions in lines 414-416: "This is the first study to show that HK L-137 supplementationfrom suckling to fattening pig might stimulate immune response and improve growth performance in the finisher stage in the same way that antibiotic growth promoters could". Firstly there were studies on immune response previously (for instance, doi: 10.3109/08923973.2012.672425); the growth performance and some immune changes in pigs also were previously investigated by authors (Watcharin Omthonglang et al., Effect of Cordyceps militaris spent mushroom substrate and Heat-killed Lactobacillus plantarum (HK L-137) supplementation in nursery pig diets on production efficiency, oxidative, and immune status). The words "in the same way" also may be incorrect if to compare action mechanism of antibiotics and HK L-137.
- Thank you for your comment. We have revised the sentence to This study shows that HK L-137 supplementation from suckling to fattening pig might stimulate immune response and promote growth performance in the finisher stage.
Reviewer 4 Report
agree.
Author Response
Many thanks once again for your comments and suggestions.
Round 3
Reviewer 2 Report
.